# Diagnostic and prognostic factors in patients with prostate cancer: a systematic review protocol

Katharina Beyer [1], Lisa Moris,[2] Michael Lardas,[3] Anna Haire,[1]
Francesco Barletta,[4] Simone Scuderi,[4] Eleni Vradi,[5] Giorgio Gandaglia,[4]
Muhammad Imran Omar,[6] Steven MacLennan [6], Jihong Zong,[7]
Bahman Farahmand,[8] Sara J Maclennan,[6] Zsuzsanna Devecseri,[9] Alex Asiimwe,[5]
Laurence Collette,[10] Anders Bjartell,[11] James Ndow,[12] Alberto Briganti,[4,13]
Mieke Van Hemelrijck [1], The PIONEER Consortium

**Correspondence to**
Katharina Beyer;
katharina.beyer@kcl.ac.uk

## ABSTRACT

**Introduction** As part of the PIONEER (Prostate Cancer Diagnosis and Treatment Enhancement Through the Power of Big Data in Europe) Consortium, we will explore which diagnostic and prognostic factors (DPFs) are currently being researched to previously defined clinical and patient-reported outcomes for prostate cancer (PCa).

**Methods and analysis** This research project will follow the following four steps: (1) a broad systematic literature review of DPFs for all stages of PCa, covering evidence from 2014 onwards; (2) discussion of systematic review findings by a multidisciplinary expert panel; (3) risk of bias assessment and applicability with Prediction model Risk Of Bias Assessment Tool criteria, Quality Assessment of Diagnostic Accuracy Studies (QUADAS-2) and the Quality In Prognosis Studies tool (QUIPS) and (4) additional quantitative assessments if required.

**Ethics and dissemination** We aim to develop an online tool to present the DPFs identified in this research and make them available across all stakeholders. There are no ethical implications.

### Strengths and limitations of this study

► This is a novel systematic review to explore relevant diagnostic and prognostic factors for previously defined clinical and patient reported prostate cancer (PCa) outcomes. It aims to increase the knowledge in this field by focusing on all available quantitative evidence for all stages of PCa.
► A multidisciplinary team including patients, urologists, oncologists, radiation oncologists, methodological experts and pathologists will be involved throughout the study.
► The search was restricted from 2014 onwards, to maintain a pragmatic approach.
► It is possible that our review will not include all articles which have been published in every journal as some may not be accessible.
► We are aware of the limitation of pooling evidence from non-randomised studies, as there could be specific bias inherent within the design. However, detailed assessments of these biases will be conducted.

## INTRODUCTION

Prostate cancer (PCa) is the second most common cancer in men worldwide[1] and accounts for 15% of all cancers diagnosed.[2] Clinically localised PCa is typically characterised by a favourable long-term natural history, where several therapeutic options are available. Moreover, the treatment landscape of advanced and metastatic PCa has changed dramatically in the past decade with the approval of multiple systemic agents, improving patients' survival. PCa is usually suspected based on the clinical findings of digital rectal examination and/or prostate-specific antigen (PSA) level.[2] However, which treatment strategy is best or which biomarkers can be used to select patients for specific therapeutic options remains largely uncertain.[3] Multiple diagnostic and prognostic factors (DPFs) are available on top of traditional PSA testing to improve the diagnosis of PCa, such as PCA3, TMPRSS2-ERG fusion, or kallikreins as incorporated in the Phi or 4Kscore test.[4–7] However, the European Association of Urology (EAU) guidelines (2019) currently do not provide recommendations to implement these factors or biomarkers into routine screening programmes due to limited data.[2]

The PIONEER (Prostate Cancer Diagnosis and Treatment Enhancement Through the Power of Big Data in Europe) Consortium is an international collaboration coordinated by the EAU which aims to establish the best evidence-based management and clinical practice of PCa across all disease stages using the power of big data analytics towards a more outcome-driven, value-based

| Workflow | Task |
|---|---|
| Stage 1. | Broad literature-based systematic review of diagnostic and prognostic factors (DPFs) for all stages of prostate cancer from 2014 onwards (English only; humans)<br>• Extract data from the included studies following the CHARMS-PF guideline |
| Stage 2. | Discussion of systematic review findings by a multidisciplinary expert panel<br>• Review the list of included studies |
| Stage 3. | Risk of Bias Assessment and applicability of individual studies using PROBAST, QUIPS and QUADAS-2 |
| Stage 4. | Quantitative assessment of individual articles using meta-analytic techniques:<br>• If PROBAST indicates low risk of bias and low concerns for applicability:<br>Oxford Classification Centre for Evidence Based Medicine:<br>1. If there is Level 1a (SR of inception cohort studies validated in different populations or Level 1 diagnostic studies from different clinical centres), we do not do a meta-analysis<br>2. No Level 1a but >2 studies which apply to 1b (Prognostic: individual inception cohort study with > 80% follow-up; CDR" validated in a single population; Diagnostic: Validating cohort study with good reference standards; or CDR" tested within one clinical centre) or 1c (Prognostic: All or none case-series; diagnostic: Absolute SpPins and SnNouts), we do a meta-analysis |
| Final aim: | Develop online PIONEER Online Search Tool for DPFs |

**Figure 1** Overview of four stage process. CDR, clinical decision rule; CHARMS PF, Critical Appraisal and Data Extraction for systematic reviews of prediction modelling; PIONEER, Prostate Cancer Diagnosis and Treatment Enhancement Through the Power of Big Data in Europe; PROBAST, Prediction model Risk Of Bias ASsessment Tool; QUADAS- 2, Quality Assessment of Diagnostic Accuracy Studies; QUIPS, Quality In Prognosis Studies tool; SR, systematic review.

and patient-centric healthcare system. A key objective is to address one of the major challenges within the context of diagnostic or prognostic biomarkers/factors: the inability to incorporate real-world clinical outcome data into the management of PCa in terms of screening, diagnosis and treatment. Biomarkers can be classified into different types: diagnostic, prognostic, predictive and therapeutic. A diagnostic biomarker or factor allows the early detection of cancer in a non-invasive way and thus the secondary prevention of cancer.[8] A prognostic biomarker or factor is a clinical or biological characteristic that provides information on the likely course of the disease.[8] In the sections below, we have used the terms biomarkers and factors interchangeably.

This project investigates which DPFs are available in relation to PIONEER's previously defined core outcomes for PCa[9–13] by evaluating at the evidence published from 2014 onwards as to reflect current medical practice and the '2014 International Society of Urological Pathology Consensus Conference on Grading Patterns and Proposal for a New Grading system'.[14] We specifically aim to assess the strength of the evidence for each DPF and use this information to develop an online search tool on the PIONEER website to be used by all stakeholders

## METHODS AND ANALYSIS

This research project will follow the following four steps (figure 1):
1. We will start with a broad systematic literature-based review of DPFs for all stages of PCa, covering English language publications of human studies from 2014 onwards.
2. The final list of all available DPFs for which a systematic review is required will then be discussed by a multidisciplinary expert panel.
3. Each study and systematic review identified through the literature search will be assessed using a risk of bias tool and if applicable the Prediction model Risk Of Bias ASsessment Tool (PROBAST) criteria.
4. For those studies with an overall low score on risk of bias and low concerns for the applicability of their results, we will use the Classification from the Oxford Centre for Evidence-Based Medicine to define whether an additional quantitative assessment is required.

### Stage 1: systematic review
The systematic review will be conducted in accordance with the Preferred Reporting Items for Systematic Reviews and Meta-Analyses guidelines.[15 16]

### Search methods and identification of studies
The literature search has been developed by an experienced Information Scientist. We will search for quantitative observational studies which assess either diagnostic or prognostic factors. The search strategy is shown in box 1.

The eligibility will be independently checked by at least two researchers. Conflicts will be solved by discussion or consulting an additional author. The abstract and full-text screening will be conducted in duplicate, and results will be compared and shared with the core group involved

**Box 1  Search strategy**

Database: OVID Medline Epub Ahead of Print, In-Process & Other Non-Indexed Citations, Ovid MEDLINE(R) Daily and Ovid MEDLINE(R) 1946 to Present, Embase <1974 to 2020 January 28>, EBM Reviews-Cochrane Database of Systematic Reviews <2005 to 21 January 2020>

**Search strategy**
1. exp *Prostatic Neoplasms/ (262435)
2. exp *prostate cancer/ (245472)
3. (prostat* adj2 (cancer* or carcinoma* or malignan* or tumor* or tumour* or neoplas* or adenocarcinoma* or adenoma*)).tw. (332251)
4. or/1–3 (366427)
5. ((diagnostic or prognos* or predict*) adj10 (biomarker or biomarkers or factor or factors)).tw,kw. (717487)
6. ((diagnostic or prognos* or predict*) adj10 (Oncotype Dx Prostate or Prolaris or Decipher or Decipher PORTOS or ProMark)).tw,kw. (458)
7. 5 or 6 (717869)
8. 4 and 7 (17456)
9. limit 8 to english language [Limit not valid in CDSR; records were retained] (16484)
10. limit 9 to yr="2014 -Current" (8417)
11. conference abstract.pt. or Congresses as Topic/ or Conference Review.pt. or "Journal: Conference Abstract".pt. (3815712)
12. 10 not 11 (5902)
13. (exp animals/ or exp animal/ or exp nonhuman/ or exp animal experiment/ or animal model/ or animal tissue/ or non human/ or (rat or rats or mice or mouse or swine or porcine or murine or sheep or lambs or pigs or piglets or rabbit or rabbits or cat or cats or dog or dogs or cattle or bovine or monkey or monkeys or trout or marmoset$1).ti.) not (humans/ or human/ or human experiment/ or (human* or men or women or patients or subjects).tw.) (10251935)
14. 12 not 13 (5882)
15. note/ or editorial/ or letter/ or Comment/ or news/ or (note or editorial or letter or Comment or news).pt. (4565255)
16. 14 not 15 (5811)
17. (child/ or Pediatrics/ or Adolescent/ or Infant/ or adolescence/ or newborn/ or (baby or babies or child or children or pediatric* or paediatric* or peadiatric* or infant* or infancy or neonat* or newborn* or new born* or adolescen* or preschool or pre-school or toddler*).tw.) not (adult/ or aged/ or (aged or adult* or elder* or senior* or men or women).tw.) (4146377)
18. 16 not 17 (5794)
19. 18 use ppez,oemezd (5788)
20. 10 use coch (6)
21. 19 or 20 (5794)
22. remove duplicates from 21 (3140)

in the research project. The EAU guidelines will also be hand-searched to identify DPFs.[2]

## Eligibility criteria
### Types of studies
We will include quantitative studies only. Studies included in this systematic review will be limited to those which are published between January 2014 and January 2020, so that the project is pragmatic. Qualitative studies, narrative literature reviews and commentary pieces will be excluded. Single studies with fewer than 50 participants will be excluded since a small patient number is unlikely to influence practice due to lack of power and

risk of selection bias in the study. In addition, the test is unlikely to be ready for clinical usage. Studies which are not published in English language or are out of our defined timeframe will also be excluded. We believe that DPFs which are relevant to clinical practice and are developed outside of the proposed timeframe will either be captured in the EAU guidelines[2] or are already included in the evidence reviewed due to high relevance.

### Types of participants
The participants are adult men (≥18 years of age) diagnosed with:
1. Localised PCa:≤T2cN0M0 with no treatment prior to their primary treatment for PCa (except for neoadjuvant androgen deprivation therapy (ADT) preceding radiotherapy). Studies in which locally advanced or metastatic disease (T3–T4 N+or M+) accounted for >10% of their participants will be excluded unless the localised participants are reported separately.
2. Locally advanced PCa: T3 or T4 and/or N1.
3. Metastatic PCa: any T, any N, M1 a-c. Studies in which local disease (≤any T, any N, M0) accounted for >10% of their participants will be excluded, unless the metastatic participants are reported separately.
4. nmCRPC: any T, N, M0 and castration-resistance defined as a castrate serum testosterone <50 ng/dL or 1.7 nmol/L plus either biochemical or radiological progression.

### Types of interventions
Interventions considered for this systematic review will be all treatments supported by the 2019 EAU guidelines[2] for localised, locally advanced, metastatic and non-metastatic castration resistant PCa.

### Data extraction
Data will be extracted by at least two reviewers and checked by a third reviewer following the Critical Appraisal and Data Extraction for systematic reviews of prediction modelling studies.[17 18]

## Stage 2: assessment of stage 1 output with expert panel to identify the individual topics for systematic reviews
An expert panel of urologists, radiologists, radiation oncologist, oncologist, methodologist and pathologists will review the extracted factors to discuss if any DPFs are missing. If no DPFs are missing, the review team will assess the quality of the identified studies for each DPF systematic review using the PROBAST criteria.[19 20]

## Stage 3: risk of bias assessment of individual articles using PROBAST
Each study and systematic review identified through the literature search will be assessed using PROBAST criteria.[19] PROBAST is a tool to assess the risk of bias as well as the applicability of diagnostic and prognostic prediction models. For studies that will not meet the PROBAST criteria, we will use Quality Assessment of Diagnostic Accuracy Studies (QUADAS-2)[21] for diagnostic factors

accuracy studies and the Quality In Prognosis Studies tool (QUIPS)[22] for prognostic factors studies.

## Stage 4: quantitative assessment of individual articles using meta-analytical techniques

We are well aware of the limitation of pooling evidence from non-randomised studies, as there could be specific bias inherent within the design. We will be very cautious while pooling evidence from non-randomised studies. For those studies with an overall low score on risk of bias and low on concerns for applicability, we will use the Classification from the Oxford Centre for Evidence-Based Medicine to define whether an additional quantitative assessment is required.

1. If there is level 1a (SR of inception cohort studies validated in different populations or level 1 diagnostic studies from different clinical centres), we do not do a meta-analysis.
2. No level 1a but >2 studies which apply to 1b (Prognostic: individual inception cohort study with >80% follow-up; clinical decision rule (CDR) validated in a single population; diagnostic: validating cohort study with good reference standards; or CDR tested within one clinical centre) or 1c (prognostic: all or none case series; diagnostic: absolute specificity is so high that a positive result rules-in the diagnosis (SpPins) and diagnostic finding whose sensitivity is so high that a Negative result rules-out the diagnosis (SnNouts)), we will perform a meta-analysis.

In the situation where we are not able to perform additional meta-analysis due to low quality of the assessed papers, we will add an additional stage to the review and discuss the data with the DPF PIONEER expert panel. We will aim to provide recommendations for researchers to improve the quality of future studies and enable the conduct of meta-analyses. We will follow the methodology developed by the Joanna Briggs Institute guidelines[23] and the framework by Arksey and O'Malley.[24]

## Patient and public Involvement

PIONEER brings together 32 key stakeholders in PCa research and clinical care from across nine countries. Consortium members are drawn from academic institutions, European organisations, patient advocacy groups, clinicians and pharmaceutical companies, as well as regulatory agencies, experts in legal data management, economics and ethics, and information and technology specialists. Hence, the patients and their family members are an integral part of all research conducted by the PIONEER Consortium.

## ETHICS AND DISSEMINATION

No additional ethical approval is required as the work will rely on published publicly available study data.

The findings of these systematic reviews will be exported into an online search tool to ensure wide applicability of the study findings. More specifically, this online search tool will produce evidence-based recommendations so that these could be used by researchers, clinicians and experts in the field. The tool will be designed so that stakeholders can access up to date available evidence (and view the quality of the studies published) when developing new DPFs or setting up clinical trials. To ensure sustainability of this tool, PIONEER and the EAU aim to update the systematic reviews described above on a regular basis to reflect the latest available research on DPFs for PCa.

### Author affiliations
[1]Translational Oncology and Urology Research (TOUR), King's College London, London, UK
[2]Department of Urology, KU Leuven University Hospitals Leuven, Leuven, Flanders, Belgium
[3]Department of Urology, Metropolitan Hospital Athens, Athens, Attike, Greece
[4]Unit of Urology/Division of Oncology, URI, IRCCS Ospedale San Raffaele, Milano, Lombardia, Italy
[5]Medical Affairs and PV, Bayer Pharma AG, Berlin, Germany
[6]Academic Urology Unit, Health Services Research Unit, University of Aberdeen, Aberdeen, UK
[7]Epidemiology, Bayer U.S, Whippany, New Jersey, USA
[8]Global Epidemiology, Bayer AG, Stockholm, Sweden
[9]Sanofi SA, Paris, Île-de-France, France
[10]EORTC, Brussels, Belgium
[11]Skåne University Hospital, Malmö, Sweden
[12]Guidelines Office, European Association of Urology, Arnhem, Netherlands
[13]Department of Urology and Division of Experimental Oncology, Urological Research Institute, Vita-Salute University San Raffaele, RCCS San Raffaele Scientific Institute, Milan, Italy

Collaborators PIONEER Consortium: Emma Smith; James N'Dow; Karin Plass; Maria Ribal; Nicolas Mottet; Lisa Moris; Michael Lardas;Thomas Van den Broeck; Peter-Paul Willemse; Giorgio Gandaglia; Karl H Pang; Riccardo Campi; Isabella Greco; Mauro Gacci; Sergio Serni (Guidelines Office European Association of Urology Arnhem Netherlands); Anders Bjartell; Susan Evans-Axelsson; Ragnar Lonnerbro (Department of Translational Medicine Lund University Lund Sweden); Alberto Briganti; Daniele Crosti; Massimiliano Meoni; Roberto Garzonio; Chris Bangma; Monique Roobol; Sebastiaan Remmers (Erasmus MC Rotterdam Netherlands); Derya Tilki (Universitätsklinikum Hamburg- Eppendorf (UKE) Hamburg Germany); Anssi Auvinen; Teemu Murtola; Tapio Visakorpi; Kirsi Talala; Teuvo Tammela; Aino Siltari (Tampere University (TAU) Tampere Finland); Mieke Van Hemelrijck; Katharina Beyer (Translational Oncology and Urology Research King's College London London); Stephane Lejeune(EORTC); Femke van Diggelen; Georgia Taxiarchopoulou; Ketharini Senthilkumar; Stefanie Schütte(TTopstart); Sophie Byrne; Luz Fialho (ICHOM); Antonella Cardone; Paulina Gono; Monica De Vetter; Klevisa Ceke (European Cancer Patient Coalition (ECPC) Brussels Belgium); Bertrand De Meulder; Charles Auffray; Irina-Afrodita Balaur; Nesrine Taibi (Imperial College London London UK); Shaun Power; Nazanin Zounemat Kermani (Association EISBM Vourles France); Kees van Bochove; Marinel Cavelaars; Maxim Moinat; Emma Voss (The Hyve Utrecht Netherlands); Chiara Bernini; Denis Horgan (European Alliance for Personalised Medicine (EAPM) Brussels Belgium); Louise Fullwood; Marc Holtorf (Pinsent Masons Leeds UK); Doron Lancet; Gabi Bernstein (Weizmann Institute Rehovot Israel); Imran Omar; Sara MacLennan; Steven MacLennan; Jemma Healey (Academic Urology Unit University of Aberdeen Aberdeen UK); Johannes Huber; Manfred Wirth; Michael Froehner; Beate Brenner; Angelika Borkowetz; Christian Thomas (Technische Universität Dresden (TUD) Dresden Germany); Friedemann Horn; Kristin Reiche; Markus Kreuz (Department of Diagnostics Fraunhofer Institute for Cell Therapy and Immunology Leipzig Germany); Andreas Josefsson; Delila Gasi Tandefelt; Jonas Hugosson (Goeteborgs Universitet (UGOT) Gothenburg Sweden); Jack Schalken; Henkjan Huisman (Radbound University Medical Center Nijmegen The Netherlands); Thomas Hofmarcher; Peter Lindgren; Emelie Andersson; Adam Fridhammar (The Swedish Institute for Health Economics (IHE) Stockholm Sweden); Alex Asiimwe; David Vizcaya; Frank Verholen; Jihong Zong; John-Edward Butler-Ransohoff; Todd Williamson; Kumari Chandrawansa; Dorothy Dlamini; Reg

Waldeck; Megan Molnar; Amanda Bruno; Ronald Herrera; Shan Jiang; Ekaterina Nevedomskaya; Samuel Fatoba; Niculae Constantinovici; Carl Steinbeißer; Sini Thomas; Monika Maass; Patrizia Torremante (Bayer AG Berlin Germany); Marc Dietrich Voss; Zsuzsanna Devecseri; Guido Cuperus (Sanofi Chilly- Mazarin France); Tom Abbott; Amit Kiran; Chad Dau Kishore; Papineni Jing; Wang-silvanto Steve; Hass Robert; Snijder Verena; Doyé Xuewei; Wang Andy Garnham (ASTELLAS); Mark Lambrecht; Russ Wolfinger; Stijn Rogiers (SAS Tervuren Belgium); Angela Servan; Florence Lefresne; Joaquin Casariego; Mohamed Samir; Joe Lawson; Katie Pascoe; Paul Robinson; Bertrand Jaton; Daniel Bakkard (Janssen Pharmaceutica NV Beerse Belgium); Heidi Turunen; Olavi Kilkku; Pasi Pohjanjousi; Olli Voima; Liina Nevalaita (Orion Corporation Espoo Finland); Christian Reich; Sonia Araujo (IQVIA London UK); Elaine Longden-Chapman; Gordon McVie; Danny Burke (The eCancer Global Foundation (eGF- eCancer) Bristol United Kingdom); Paul Agapow; Sahra Derkits; Muriel Licour; Charles McCrea; Sarah Payne; Alan Yong; Lucy Thompson; Flavia Lujan (AstraZeneca); Billy Franks (Julius Clinical); Michael Bussmann; Inken Köhler (HelmHoltz).

**Contributors** Design of protocol: LM, ML, AH, FB, SS, EV, GG, MIO, SM, JZ, BF, SJM, ZD, AA, LC, ABj, JN, ABr and MVH. Draft of manuscript: KB, MIO, JN and MVH. Final approval of manuscript: LM, ML, AH, FB, SS, EV, GG, MIO, SM, JZ, BF, SJM, ZD, AA, LC, ABj, JN, ABr and MVH.

**Funding** PIONEER is funded through the IMI2 Joint Undertaking and is listed under grant agreement No. 777492. IMI2 receives support from the European Union's Horizon 2020 research and innovation programme and the European Federation of Pharmaceutical Industries and Associations (EFPIA).

**Competing interests** None declared.

**Patient and public involvement** Patients and/or the public were involved in the design, or conduct, or reporting, or dissemination plans of this research. Refer to the Methods section for further details.

**Patient consent for publication** Not required.

**Provenance and peer review** Not commissioned; externally peer reviewed.

**ORCID iDs**
Katharina Beyer http://orcid.org/0000-0002-8450-8850
Steven MacLennan http://orcid.org/0000-0002-2691-8421
Mieke Van Hemelrijck http://orcid.org/0000-0002-7317-0858

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
