## [Reviewer comments · BMJ Open]

ARTICLE DETAILS

TITLE (PROVISIONAL)	A systematic review protocol of diagnostic and prognostic factors in patients with prostate cancer
AUTHORS	Beyer, Katharina; Moris, Lisa; Lardas, Michael; Haire, Anna; Barletta, Francesco; Scuderi, Simone; Vradi, Eleni; Gandaglia, Giorgio; Omar, Muhammad; MacLennan, Steven; Zong, Jihong; Farahmand, Bahman; MacLennan, Sara; Devecseri, Zsuzsanna; Asimwe, Alex; Collette, Laurence; Bjartell, Anders; NDow, James; Briganti, A; Van Hemelrijck, Mieke

VERSION 1 – REVIEW

REVIEWER	Bhavina Batukbhai Sharma (Bhavina Batukbhai) Dartmouth Hitchcock Medical Center, USA
REVIEW RETURNED	08-Aug-2020

GENERAL COMMENTS	The authors have described a protocol for a systematic review to assess the availability of different diagnostic and prognostic factors for patients with prostate cancer. Overall, this is a well written protocol with few minor concerns listed below. The authors have appropriately highlighted the lack of validated biomarkers in the standard of care for prostate cancer and the need to create an evidence database to educate future use of the available biomarkers. Recommendations: - In the introduction section: Please include ' in men' after 'prostate cancer is the second most common cancer' (as in men and women combined, it is breast cancer)- Please elaborate the aim of the systematic review (i.e is to assess strength of evidence with each DPF and to use this to develop an online search tool...) and define what is meant by the ' previously defined outcomes'- In the Methods section: please clarify what is meant by 'interventions' (is it treatment or diagnostic procedures...)- Please include details on analytic approach for the data derived in this systematic review (i.e how will the data be presented (relative risk, odds ratio...), how will the authors deal with missing or incomplete data, any possible subgroup or sensitivity analysis...)
--

REVIEWER	Caroline M Moore University College London, UK
REVIEW RETURNED	20-Oct-2020

GENERAL COMMENTS	This protocol for a systematic review outlines a robust process for assessing the published evidence on diagnostic and prognostic factors in patients with prostate cancer.
---

	The assembled team have the appropriate range of expertise and are taken from the PIONEER consortium. Minor comment: 1. The resolution of conflict (p3) should say that conflicts will be solved by discussion or by consulting an additional author, rather than third party as three will have already reviewed it.
REVIEWER	Pepe Pietro Cannizzaro Hospital - Urology Unit Catania (Italy).
REVIEW RETURNED	20-Oct-2020
GENERAL COMMENTS	The Protocol is interesting and well written.

VERSION 1 – AUTHOR RESPONSE

Reviewer: 1
Reviewer Name

Bhavina Batukbhai Sharma (Bhavina Batukbhai), Institution and Country, Dartmouth Hitchcock Medical Center, USA.

Comments to the Author

The authors have described a protocol for a systematic review to assess the availability of different diagnostic and prognostic factors for patients with prostate cancer. Overall, this is a well written protocol with few minor concerns listed below. The authors have appropriately highlighted the lack of validated biomarkers in the standard of care for prostate cancer and the need to create an evidence database to educate future use of the available biomarkers.

Recommendations:

- In the introduction section: Please include ' in men' after 'prostate cancer is the second most common cancer' (as in men and women combined, it is breast cancer)

Reply

This has been included in the revised manuscript.

“Prostate cancer (PCa) is the second most common cancer in men worldwide 1 and accounts for 15% of all cancers diagnosed 2.”

- Please elaborate the aim of the systematic review (i.e is to assess strength of evidence with each DPF and to use this to develop an online search tool...) and define what is meant by the ' previously defined outcomes'

Reply

We included the following elaboration in the manuscript.

“This project aims to investigate which DPFs are available in relation to PIONEER’s previously defined core outcomes for PCa 9-13 by evaluating the evidence published from 2014 onwards - as to reflect current medical practice and the ‘2014 International Society of Urological Pathology (ISUP) Consensus Conference on Grading Patterns and Proposal for a New Grading system’14. We specifically aim to assess the strength of the evidence for each DPF and use this information to

develop an online search tool on the PIONEER website to be used by all stakeholders.”

- In the Methods section: please clarify what is meant by 'interventions' (is it treatment or diagnostic procedures...)

Reply:

The following was added to clarify intervention.

“Types of interventions: Interventions considered for this systematic review will be all treatments supported by the 2019 EAU guidelines for localised, locally advanced, metastatic and non-metastatic castration resistant PCa.”

- Please include details on analytic approach for the data derived in this systematic review (i.e how will the data be presented (relative risk, odds ratio...), how will the authors deal with missing or incomplete data, any possible subgroup or sensitivity analysis...)

Reply

Any quantification will be used depending on what information is available in the identified literature. Due to the heterogeneity and rather low quality of data available (also highlighted by Eggenner et al, 2019,) we are not able to define this a priori. We have, however, started our assessments and therefore already discussed the possibility of presenting the data as a scoping review. We have hence added the following to our protocol.

“In the situation where we are not able to perform additional meta-analysis due to low quality of the assessed papers, we will add an additional stage to the review and discuss the data with the DPF PIONEER expert panel. We will aim to provide recommendations for researchers to improve the quality of future studies and enable the conduct of meta-analyses. We will follow the methodology developed by the Joanna Briggs Institute guidelines 23 and the framework by Arksey and O'Malley 24.”

Reviewer: 2

Reviewer Name

Caroline M Moore, Institution and Country, University College London, UK

Please state any competing interests or state 'None declared':

None declared

Comments to the Author

This protocol for a systematic review outlines a robust process for assessing the published evidence on diagnostic and prognostic factors in patients with prostate cancer.

The assembled team have the appropriate range of expertise and are taken from the PIONEER consortium.

Minor comment:

1. The resolution of conflict (p3) should say that conflicts will be solved by discussion or by consulting an additional author, rather than third party as three will have already reviewed it.

Reply

Thank you, we have amended this in the revised manuscript.

“The eligibility will be independently checked by at least two researchers. Conflicts will be solved by discussion or consulting an additional author.”

Reviewer: 3

Comments to the Author

No comment